# Prevalence and Characterisation of Multiresistant Bacterial Strains Isolated in Pigs from the Island of Tenerife

**DOI:** 10.3390/vetsci9060269

**Published:** 2022-06-03

**Authors:** Rossana Abreu, Cristobalina Rodríguez-Álvarez, Beatriz Castro-Hernandez, Maria Lecuona-Fernández, Juan Carlos González, Yurena Rodríguez-Novo, Maria de los Angeles Arias Rodríguez

**Affiliations:** 1Department of Preventive Medicine and Public Health, University of La Laguna, Campus de Ofra, s/n, 38071 Santa Cruz de Tenerife, Spain; rabreu@ull.edu.es (R.A.); crrodrig@ull.edu.es (C.R.-Á.); 2Microbiology and Infection Control Service, University Hospital of the Canary Islands, Canary Islands, Tenerife, 38320 San Cristóbal de La Laguna, Spain; mbcasher@gobiernodecanarias.org (B.C.-H.); mlecfer@gobiernodecanarias.org (M.L.-F.); 3Canary Islands Health Service, Canary Islands, 38004 Santa Cruz de Tenerife, Spain; ecoladera@gmail.com; 4Faculty of Health Sciences, Nursing Section, University of La Laguna, Tenerife, 38200 La Laguna, Spain; yrodrign@ull.es

**Keywords:** pig livestock, antimicrobial resistance, one-health

## Abstract

Background: Antibiotic-resistant bacteria can circulate among human and animal populations through direct contact with animals, as well as via food and the environment. The purpose of this study was to examine the prevalence and characterisation of multiresistant bacteria in pig samples. Methods: 224 samples of pig livestock were taken at the slaughterhouse on the island of Tenerife. A nasal and a rectal sample were collected from each pig. The presence of methicillin-resistant *Staphylococcus aureus* (MRSA), methicillin-resistant *Staphylococcus* coagulase-negative (MRCoNS), vancomycin-resistant Enterococcus (VRE), extended-spectrum ß-lactamase-producing *Enterobacteriaceae* (BLEE), carbapenemase-producing *Enterobacteriaceae* (CPE), and colistin-resistant Enterobacteriaceae was investigated. The resistance genes of the isolated bacteria were characterised by specific PCRs depending on the microorganism to be studied, and in vitro antimicrobial resistance was determined using the broth microdilution method (Vitek^®^2 system bioMérieux^®^, Nurtingen, Germany). Results: MRSA prevalence was 73.21% (164 isolates). MRCoNS prevalence was 9.8% (22 isolates), *S. sciuri* being the prevalent species. Six isolates presented a 2.7% prevalence of extended-spectrum ß-lactamase-producing *Escherichia coli* (BLEE) in the CTX-M-1 group. No vancomycin-resistant *Enterococcus* (VRE), carbapenemase-producing *Enterobacteriaceae* (CRE), or colistin-resistant *Enterobacteriaceae* were isolated. Conclusion: we found a high presence of multiresistant bacteria, suggesting the need for increased control and surveillance of this type of strains in pig livestock and a better understanding of the possible transmission routes of these microorganisms through livestock products.

## 1. Introduction

Antibiotic resistance is a global health concern that has been linked to humans, animals, and environmental factors [1,2]. The persistence and spread of resistant microbial species and the association of determinants at the human–animal–environment interface can alter microbial genomes, resulting in resistant superbugs in various niches [2].

The term One Health recognises the importance of increasing interdisciplinary collaboration in human, animal, and environmental healthcare to enable the creation and implementation of programmes, policies, and legislation to improve public health [2,3,4]. The usage of antibiotics, persistence of antibiotic residues, and presence of resistant bacteria in the human–animal–environment niches are associated with the One Health triad due to the interdependence of these pillars in the food chain and environment [2].

The issue of emerging resistant microorganisms associated with livestock is closely linked to improper use of antimicrobial agents in veterinary care, as well as to international trade of food of animal origin, which can contribute to the spread of resistant strains [1,5].

Drug-resistant bacteria can circulate among human and animal populations via food, water, and the environment. This transmission is influenced by trade, travel, human migration, and transhumance [6,7]. Although antibiotic consumption in human and animal care has decreased within the European Union in the past few years, it remains high [8].

Staphylococcus *aureus* is a pathogen that develops resistance to multiple antimicrobial agents, particularly to methicillin (MRSA). It is a significant cause of hospital-acquired infections (HA-MRSA), as well as community-acquired infections (CA-MRSA) and livestock-associated infections (LA-MRSA) [9,10,11,12]. Methillicin-resistant coagulase-negative Staphylococcus (MRCoNS) has often been isolated in food-producing animals, as well as in meat [13,14,15].

The mecA gene, responsible for methicillin resistance, is found in SCCmec cassettes, which are the mobile genetic elements responsible for the transfer of resistance genes [16]. These genes can be part of the genome in Staphylococcus species, such as *S. aureus* and coagulase-negative Staphylococcus [7,17].

Vancomycin-resistant Enterococcus (VRE) is among the first antibiotic-resistant bacteria documented whose primary origin is animal farming [18,19], and there are various studies on the detection of these multiresistant bacteria in swine livestock [20,21].

Extended-spectrum ß-lactamase producing Enterobacteriaceae (BLEE) are often isolated in clinics [22] and animals, particularly chicken [23], and they have also been detected in other farm animals and their meat products [5,24].

Colistin has been widely used orally since the 1960s in farm animals, especially pigs, to control infections by Enterobacteria, leading to the emergence of resistant strains in humans and animals [25,26].

Pigs represent the largest livestock category in the European Union. In 2020, there were 146 million pigs in the EU. The two main pork producing Member States are Germany (5.1 million tonnes in 2020) and Spain (5.0 million tonnes) [27].

The presence of multiresistant bacteria in this type of livestock is a major public health concern. The purpose of this study was to investigate the presence of methicillin-resistant Staphylococcus (MRSA), vancomycin-resistant Enterococcus (VRE), extended-spectrum ß-lactamase-producing Enterobacteriaceae (BLEE), carbapenemase-producing Enterobacteriaceae (CRE), and colistin-resistant Enterobacteriaceae (CRE) in pig samples of the island of Tenerife, as well as to determine their phenotypic and genotypic characteristics.

## 2. Materials and Methods

### 2.1. Samples

A cross-sectional prevalence study was conducted. A total of 224 pigs were examined. A random selection of 12 wean-to-finish farms for local consumption within the 39 existing on the island was previously performed. The number of pigs to sample on each farm was selected according to its size. They all were intensive farms representative of the standard pig farms of the island, according to the veterinarians of the Canary Islands Health Service

Pig samples were collected at the Island Slaughterhouse of Tenerife between September 2020 and January 2021. Animals were transported by farm truck to the slaughterhouse, where they were kept in separate stables, according to their farm of origin, and slaughtered within 12 h. Transportation time ranged from 1 to 2 h.

The samples were taken by a veterinarian of the slaughterhouse just before slaughter and while the animals were unconscious. Nasal and rectal swabs preserved with AmiesRayon (Deltalab^®^) were taken and subsequently transported to the lab at room temperature until processed.

### 2.2. Isolation and Identification

In the case of MRSA and MrCoNS, nasal samples were processed. However, for the remaining bacteria studied, rectal exudate was used.

Direct seeding of the nasal swabs was performed in chromID^®^MRSA SMART agar (bioMérieux^®^, Nurtingen, Germany), as well as indirect seeding by incubation in a brain-heart infusion broth (BHI) with NaCl at 6.5% in an Eppendorf tube for 18–24 h at 37 °C. Subsequently, 10 μL of this infusion broth was inoculated in chromID^®^MRSA SMART agar. Direct seeding of the rectal swabs was performed in agar MacConkey, ChromID™ESBL agar, ChromID^®^CARBA SMART (bioMérieux^®^) agar, to be finally introduced in an Eppendorf tube with 2–3 mL of brain–heart infusion broth (BHI) and a concentration of vancomycin of 3.3 mg/L at 37 °C/24 h, to be subsequently seeded in agar chromID™VRE (BioMérieux^®^). Culture media were incubated at 37 °C for 24–48 h. Multidrug-resistant (MDR) organisms were defined as acquired non-susceptibility to at least one agent in three or more antimicrobial categories.

#### 2.2.1. Methicillin-Resistant Staphylococcus (MRS)

*Staphylococcus* suspicious colonies were identified by morphology and growth colour. Species identifications were confirmed by Vitek II Automated Microbiology System (bioMérieux^®^). Methicillin resistance was confirmed by detecting the presence of penicillin-binding protein A (PBP2a) (MRSA screen, Denka Seiken Co™, Tokio, Japón), and mecA gene presence was confirmed by real-time PCR (IQ ™ 5; Rad, Hércules, CA, USA, EE. UU.) with the primers mecA- 1: GGG GTG GTT ACA ACG TTA CAA G and mecA- 2: AGT TCT GCA GTA CCG GAT TTG C (95 °C, 5 min, (94 °C/60 °C/72 °C) 1 min × 30 cycles). Clonal relationships were studied using pulsed-field gel electrophoresis (PFGE). Macrorestriction was performed using the ApaI enzyme (Promega). Results were interpreted according to the criteria described by Tenover et al. [28]. All MRSA isolates were analysed using MLST, as described by Enright et al. [29]. To extract bacterial DNA, a DNeasy Blood and Tissue Kit (Qiagen, EE.UU) was used. Allelic profiles and sequence types were assigned according to the *S. aureus* MLST database [30]. In vitro antimicrobial resistance was determined using the broth microdilution method (Vitek^®^2 system bioMérieux^®^). The antibiotics tested were as follows: benzylpenicillin, oxacillin, gentamicin, tobramycin, levofloxacin, erythromycin, clindamycin, linezolid, daptomycin, teicoplanin, vancomycin, tigecycline, fosfomycin, fusidic acid, mupirocin, rifampicin, and trimethoprim-sulfamethoxazole (SXT). *S. aureus* ATCC 29213 was used as a reference strain, and all procedures and test interpretations followed the Clinical and Laboratory Standards Institute (CLSI) guidelines [31].

#### 2.2.2. Vancomycin-Resistant Enterococcus (VRE)

VRE colonies were preliminarily identified in the culture medium, the colonies presenting a violet colour (*E. faecium*) or those showing a blue-green colour (*E. faecalis*). Species identifications were confirmed by Vitek^®^2 (bioMérieux^®^) system. The enzyme used for macrorestriction was SmaI (Promega). Results were interpreted according to the criteria described by Tenover et al. [28]. Allelic profiles and sequence types were assigned according to the public databases for molecular typing and microbial genome diversity (MLST) MLST (http://www.mlst.net, accessed on 20 October 2020). In vitro antimicrobial resistance was determined using the broth microdilution method (Vitek^®^2, bioMérieux^®^). The antibiotics tested were as follows: benzylpenicillin, oxacillin, gentamicin, tobramycin, levofloxacin, erythromycin, clindamycin, linezolid, daptomycin, teicoplanin, vancomycin, tigecycline, nitrofurantoin, and trimethoprim-sulfamethoxazole (SXT). All procedures and test interpretations followed the Clinical and Laboratory Standards Institute (CLSI) guidelines [31].

#### 2.2.3. Extended-Spectrum ß-Lactamase-Producing Enterobacteriaceae (ESBL)

Colonies presenting a spontaneous colouration between pink and burgundy, characteristic of β-glucuronidase-producing strains, in ChromID™ESBL agar medium and confirmed by the Vitek^®^2 (bioMérieux^®^) system, were interpreted as positive for *E. coli* ESBL. ESBL producers were confirmed by the double-disk synergy test, using both cefotaxime and ceftazidime alone and in combination with clavulanic disks. *E. coli* ATCC 25922 was used as a reference strain. Clonal relationships were studied using pulsed-field gel electrophoresis (PFGE), and macrorestriction was performed using XbaI (Promega) enzyme. Results were interpreted according to the criteria described by Tenover et al. [28]. The presence of CTX-M-type ESBL was detected by real-time polymerase chain reaction (RT-PCR) using the Kit RealCycler BLACTX-U (RealCycler, Progenie Molecular, Valencia, Spain^®^) to detect CTX-M-1. Group Nucleic acid purification was performed with Maxwell TM 16 viral total nucleic acid purification kit (Promega^®^). In vitro antimicrobial resistance was determined using the broth microdilution method by Vitek^®^ 2, bioMérieux^®^ system. The antibiotics tested were as follows: amoxicillin/clavulanic acid, ampicillin, cefuroxime, axetil, cefoxitin, cefotaxime, ceftazidime, cefepime, ertapenem, imipenem, amikacin, gentamicin, nalidixic acid, ciprofloxacine, tigecycline, and trimethoprim-sulfamethoxazole. All procedures and test interpretations followed the Clinical and Laboratory Standards Institute (CLSI) guidelines [31].

#### 2.2.4. Carbapenemase-Producing Enterobacteriaceae (CPE)

Colonies presenting a characteristic spontaneous colouration between pink and burgundy in ChromID^®^CARBA SMART (bioMérieux^®^) agar culture medium were interpreted as positive for carbapenemase-producing *Enterobacteriaceae*. Species identifications were confirmed by Vitek^®^2 (bioMérieux^®^) system. Minimal inhibitory concentration (MIC) was determined using the broth microdilution method described by the Clinical and Laboratory Standards Institute [30]. The carbapenemic antibiotics included in the study were as follows: imipenem, meropenem, ertapenem, and doripenem.

#### 2.2.5. Colistin-Resistant Enterobacteriaceae

Colistin MIC determination was performed using broth microdilution by Etest–Colistin (Liofilchem ^®^ Mic Test Strip, Italy). It was interpreted using the cut-off points set by the European Committee on Antimicrobial Susceptibility Testing (sensitive ≤ 2 mg/mL, resistant > 2 mg/mL) [32].

## 3. Results

The presence of vancomycin-resistant *Enterococcus* or colistin-resistant and carbapenemic-resistant *Enterobacteriaceae* was not detected in this study.

Out of the 224 nasal samples collected, 164 were found positive for MRSA, representing a prevalence percentage of 73.21%. Pulsed-field gel electrophoresis (PFGE) proved most isolates were highly related, and Multilocus sequence typing (MLST) confirmed all isolates belonged to Sequence Type 398 (ST398).

In the study, 22 isolates of coagulase-negative methicillin-resistant *Staphylococcus* were detected, with a positivity rate of 9.8%. Identified species are shown in Table 1.

Table 2 shows the antibiotic resistance of MRSA isolates to the antibiotics tested. In the case of MRSA isolates, the highest resistance rates corresponded to clindamycin and aminoglycosides, as well as to non-carbapenem β-lactam antibiotics, while in the case of MRCoNS, they corresponded to clindamycin and fosfomycin.

The resistance patterns of methicillin-resistant *Staphylococcus* spp. strains are shown in Table 3. In the case of MRSA, 13 different resistance patterns were found, of which resistance to non-carbapenem β-lactam antibiotics, as well as to the aminoglycosides tested, clindamycin, and clotrimazole, were found to be predominant. In the case of MRCoNS, four different resistance patterns were obtained, of which resistance to non-carbapenem β-lactam antibiotics, as well as to erythromycin, clindamycin, cotrimoxazole, and fosfomycin were found to be predominant.

One hundred per cent of our identified MRSA and MRCoNS strains were multiresistant, presenting resistance to at least one antibiotic in three of the groups tested.

In six rectal samples, extended-spectrum ß-lactamase-producing E. coli (BLEE) were isolated, all corresponding to the bla CTX-M1 group. The resistance pattern for all E. coli bla CTX-M1 isolates was against non-carbapenem ß-lactam antibiotics and cotrimoxazole. Positive samples came from two farms in the south of Tenerife.

## 4. Discussion

A high presence of multiresistant bacteria was found in the pigs sampled, mainly MRSA strains. In the case of MRSA and MrCoNS, nasal samples were collected since most studies on animals and humans indicate that nasal samples are the best to isolate these bacteria [7,11]. However, for the remaining bacteria studied, rectal exudate was used. They all came from intensive farms, and the samples were taken at the slaughterhouse, both factors which can contribute, according to various studies, to such a high presence of resistant strains [1,10,33,34].

MRSA prevalence in our study is higher than that obtained by some authors [7,11,35] and similar to that found by Sahibzada et al. [36] and Pirolo et al. [37].

In a study we performed ten years ago, an MRSA prevalence of 85.7% was obtained [10], higher than the current percentage of 73.21%. Increased antibiotic control by health authorities nowadays is likely to have slightly diminished the prevalence of these strains. However, a wide presence of these strains remains in our habitat. Dierikx et al. [38] found a very high prevalence of MRSA nasal carriers in Dutch slaughter pigs, comparing a 10- year period and indicating the decrease in the use of antimicrobials in the Netherlands had not had an effect on the rate of MRSA carriers in pigs during the study period.

In our study, all isolates belonged to Sequence Type 398 (ST398). LA-MRSA CC398 strain has been described in many countries, in Europe and worldwide, with different prevalence rates, in pig livestock and workers of the sector [11,15,36,37].

We found a significant presence of MRSA multiresistant strains, with a high percentage of resistance to aminoglycosides (gentamicin (59.8%), tobramycin (62.2%)), unlike Sahibzada et al. [36], who found that all MRSA isolates from pigs were sensitive to gentamicin, but obtained a higher resistance to clindamycin (97.6%,) and erythromycin (96.3%), which, in our study, were 79.3% and 18.3%, respectively.

In our study, it is worth noting the presence of methicillin-resistant coagulase-negative Staphylococcus spp. (MRCoNS), which are often isolated in pigs and can be an important reservoir of resistant genes [7,38,39]. The most identified species in our study was S. sciuri (54,5%), similarly to Bonvegna et al. [7], based on Italian farm pigs. However, Vanderhaeghen et al. [40] identified Staphylococcus epidermidis (38.9%), followed by Staphylococcus sciuri (18.1%), and other species in a smaller percentage, in pigs from Belgian farms.

As to the resistance of these strains to other non-β-lactam antibiotics, differences were observed between MRSA and MRCoNS strains. In the case of MRSA strains, the highest resistances were obtained against aminoglycosides, yet all MRCoNS isolates were sensitive. Both *Staphylococcus* groups were highly resistant to cotrimoxazole and clindamycin. All isolates showed high resistance to fosfomycin, although in a much higher percentage in the case of SCoNRM (81.8%). These presented a high resistance to fusidic acid (63.6%), which was not found in MRSA. Resistance patterns were different. Thus, 50% of MRSA strains showed resistance against non-carpabenem β-lactam antibiotics, as well as against aminoglycosides, clindamycin, and cotrimoxazole, whereas 40.9% of MRCoNS strains presented resistance to ß-lactam antibiotics, as well as to erythromycin, clindamycin, fosfomycin, and cotrimoxazole.

In our study, we detected, in a low percentage, the presence of extended-spectrum ß-lactamase-producing *Escherichia coli* strains, which had not previously been detected in pigs in Tenerife, although they had been found in chicken [23]. This could point to the existence of a new reservoir of these bacteria in pigs, in agreement with other studies [40,41,42]. CTXM-1 genes were detected in 100% of isolated strains, unlike Bernreiter-Hojer, who reported TEM-1 (56%) and CTXM-1. (13.71%) to be the most predominant β-lactamase gene classes [43].

The presence of vancomycin-resistant *Enterococcus* (VRE), as well as carbapenemase-producing and colistin-resistant Enterobacteriaceae (CRE), was not found, despite various studies describing the presence of these multiresistant strains in pig livestock [44,45].

The limitations of our study were that the swine included always came from intensive farms whose information on antimicrobial use was not available to us and that environmental samples from the farms and the slaughterhouse were not included. Most antibiotics studied are not normally used in pig livestock, except quinolones and aminoglycosides. However, we tested clinically used antibiotics since our main concern was to find out whether these strains of animal origin could show resistance to antibiotics used in human clinical practice, which are even sometimes restricted to hospital use.

Aslam et al. [2] point to animal health and bacterial resistances of strains of animal origin as one of the links contributing to the emergence of superbugs. However, the real contribution of these strains is not known exactly. The first report on the “Study of the main sources of emission, dispersion routes, and ways of exposure to antimicrobials, resistant bacteria, and antimicrobial resistance genes for humans and animals” has recently been published in Spain, within the national programme of surveillance of resistance to antibiotics. It emphasises the relevance of livestock production as a source of emission of antibiotic substances and its related Public Health concerns, as well as highlights the need to continue working on the One Health programme [46]. At a European level, the JIACRA report concludes that the reduction of antibiotic consumption would have the biggest impact on the corresponding population, human or animal, but it also states that the reduction of antibiotic use in veterinarian medicine would also have a positive impact, although to a lesser extent, on the percentages of resistance of bacterial isolates from humans [47].

All pigs included in our study came from intensive livestock farming, where treatment and prevention of animal infections are often performed collectively rather than individually. We believe that this strategy is one of the most harmful practices leading to the development of resistance and therefore needs to be eliminated since antibiotics should not be used to compensate for lack of hygiene, biosafety, or proper animal handling.

## 5. Conclusions

We found a high presence of multiresistant bacteria, suggesting the need for increased control and surveillance of this type of strains in pig livestock and a better understanding of the possible transmission routes of these microorganisms through livestock products. Monitoring the presence of resistant bacteria in the human–animal–environment niches is essential to achieving the One Health objective.

## Figures and Tables

**Table 1 vetsci-09-00269-t001:** Prevalence of coagulase-negative methicillin-resistant Staphylococcus species.

Microorganism	Positive Samples No (%)
*Staphylococcus sciuri*	12 (54.5)
*Staphylococcus haemolyticus*	4 (18.2)
*Staphylococcus lentus*	2 (9.1)
*Staphylococcus gallinarum*	2 (9.1)
*Staphylococcus warneri*	2 (9.1)

**Table 2 vetsci-09-00269-t002:** Percentage of resistance to the antibiotics tested presented by the isolated MRS strains.

Antibiotic	MRSA	MRCoNS
	Resistant No (%)	Resistant No (%)
Benzylpenicillin	164 (100)	22 (100)
Oxacillin	164 (100)	22 (100)
Gentamicin	98 (59.8)	0 (0)
Tobramycin	102 (62.2)	0 (0)
Levofloxacin	16 (9.8)	2 (9,1)
Erythromycin	30 (18.3)	0 (0)
Clindamycin	130 (79.3)	16 (72.7)
Linezolid	0 (0)	0 (0)
Daptomycin	0 (0)	0 (0)
Teicoplanin	0 (0)	0 (0)
Vancomycin	0 (0)	0 (0)
Tigecycline	22 (13.4)	0 (0)
Fosfomycin	72 (43.9)	18 (81.8)
Fusidic acid	0 (0)	14 (63.6)
Mupirocin	0 (0)	0 (0)
Rifampicin	0 (0)	0 (0)
Cotrimoxazole	100 (60.9)	5 (22.7)

**Table 3 vetsci-09-00269-t003:** Resistance patterns of methicillin-resistant *Staphylococcus* spp. strains.

Microorganisms	Resistance Pattern *	Number
MRSA	PG + OXA + GM + TM + CC + STX	82
MRSA	PG + OXA + CC + FM + NI	32
MRSA	PG + OXA + LV + E + TGC	12
MRSA	PG + OXA + GM + TM + STX	6
MRSA	PG + OXA + GM + TM + CC	6
MRSA	PG + OXA + E + TGC + FM	6
MRSA	PG + OXA + E + CC + FM + STX	6
MRSA	PG + OXA + E + TGC + FM + STX	4
MRSA	PG + OXA + GM + TM	2
MRSA	PG + OXA + GM + TM + CC + FM	2
MRSA	PG + OXA + E + CC + FM	2
MRSA	PG + OXA + TM + LV	2
MRSA	PG + OXA + TM + LV + STX	2
MRCoNS	PG + OXA + E + CC+ FM + STX	9
MRCoNS	PG + OXA + CC + E + FM	8
MRCoNS	PG + OXA + E + CC+ FM + FA + STX	3
MRCoNS	PG + OXA + E + FM	2

* PG—benzylpenicillin; OXA—oxacillin; GM—gentamicin; TM—tobramycin; E—erythromycin; CC—clindamycin; FM—fosfomycin; NI—nitrofurantoin; LVX—levofloxacin; TGC—tigecycline; SXT—trimethoprim/sulphamethoxazole.

## Data Availability

The data presented in this study are available on request from the corresponding author.

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
