# Peer review of "Prevalence and Characterisation of Multiresistant Bacterial Strains Isolated in Pigs from the Island of Tenerife"

_vetsci, 2022, doi:10.3390/vetsci9060269_

Round 1
Reviewer 1 Report
The authors present a study regarding the isolation and characterization of multiresistant bacterial strains from nasal and rectal swabs of pigs from the island of Tenerife.
Overall, the authors focus on the One Health framework, highlighting the impact of veterinary use of antimicrobials, among other factors, in the emergence of multidrug resistant bacteria, specifically the role of pigs as source of those pathogens, namely Methicillin Resistant Staphylococcus aureus, Methicillin Resistant coagulase negative Staphylococcus, Vancomycin-resistant Enterococcus and extended-spectrum β-lactamase producing Enterobacteriaceae, carbapenemase-producing Enterobacteriaceae, and colistin-resistant Enterobacteriaceae.
The study presented brings valuable information regarding the presence of multiresistant bacteria in pigs and their possible role in the dissemination of those pathogens.
Nevertheless, some questions arise and should be clarified:
- The sampling procedure is not very clear. How many pigs from each farm were sampled? Do these 224 animals represent the pig production in Tenerife? Were they all fattening pigs or where there any sows or adult boars? Since fattening pigs have a shorter productive live when compared with sows and adult boars can that influence any of the results?
- The PCR procedures are not clear. Authors should provide primer sequences, cycling conditions or a reference for those.
- What was the criterion used for defining an isolate as multiresistant?
- Some information regardign the results are not clearly stated, namely the relative prevalence of multiresistant MRSA regarding the overall MRSA isolated. Is it 160/164? The authors could provide that information in a table collapsing all the results.
- What is the incidence of human cases of LA-MRSA CC398 infection in Spain and in particular in Tenerife? Are they associated with pork or with direct contact with live pigs? Could humans working int those pig farms be carriers of those MRSA? What it is the role of pork in this scenario? It would be interesting if this issue could be discussed in the One Health perspective.
- The authors state that information regarding the antimicrobial use in pig farms was not available. Despite this limitation, can all of the antimicrobials tested in vitro be used in pig production? It would be interesting if this issue could be discussed.
- The authors suggest increased control and surveillance of multiresistant bacteria in pig livestock. Considering that every country of the EU should monitor the antimicrobial resistance in bacteria from livestock, what additional measures should be enforced? It would be interesting if this issue could be discussed.
Author Response
Thank you very much for the comments and time spent reading this work.
The authors present a study regarding the isolation and characterization of multiresistant bacterial strains from nasal and rectal swabs of pigs from the island of Tenerife.
Overall, the authors focus on the One Health framework, highlighting the impact of veterinary use of antimicrobials, among other factors, in the emergence of multidrug resistant bacteria, specifically the role of pigs as source of those pathogens, namely Methicillin Resistant Staphylococcus aureus, Methicillin Resistant coagulase negative Staphylococcus, Vancomycin-resistant Enterococcus and extended-spectrum β-lactamase producing Enterobacteriaceae, carbapenemase-producing Enterobacteriaceae, and colistin-resistant Enterobacteriaceae.
The study presented brings valuable information regarding the presence of multiresistant bacteria in pigs and their possible role in the dissemination of those pathogens.
Nevertheless, some questions arise and should be clarified:
- The sampling procedure is not very clear. How many pigs from each farm were sampled? Do these 224 animals represent the pig production in Tenerife? Were they all fattening pigs or where there any sows or adult boars? Since fattening pigs have a shorter productive live when compared with sows and adult boars can that influence any of the results?
We have included the clarification of the sampling procedure in the material and methods section. All pigs were fattening pigs.
You are right. Indeed, there may be a difference in colonisation by resistant bacteria, as sows and adult boars spent longer time in the farm than fattening pigs, but we have not studied that point.
- The PCR procedures are not clear. Authors should provide primer sequences, cycling conditions or a reference for those.
This has already been modified in the material and methods section and we thus believe it has been clarified
- What was the criterion used for defining an isolate as multiresistant?
This has already been added in the material and methods section
MDR was defined as acquired non-susceptibility to, at least, one agent in three or more antimicrobial categories.
- Some information regardign the results are not clearly stated, namely the relative prevalence of multiresistant MRSA regarding the overall MRSA isolated. Is it 160/164? The authors could provide that information in a table collapsing all the results.
According to the definition of multiresistance used, which has been included in the material and methods section, all our Staphylococcus spp. isolates were multiresistant (164). These results have been added in the corresponding section.
- What is the incidence of human cases of LA-MRSA CC398 infection in Spain and in particular in Tenerife? Are they associated with pork or with direct contact with live pigs? Could humans working int those pig farms be carriers of those MRSA? What it is the role of pork in this scenario? It would be interesting if this issue could be discussed in the One Health perspective.
We have not carried out studies to detect resistant bacteria using pork samples. As to the prevalence in carriers within staff working in contact with pigs in farms or slaughterhouses, in a previous study performed by our research team we found 9.3% of carriers of this animal strain (Morcillo et al. 2012). Our study does not include data on prevalence of these strains in human infections. However, it would be important to carry out a joint study on determination and characterization of resistant strains covering One-Health’s three links: humans, animals, and the environment.
The first report on the “Study of the main sources of emission, dispersion routes and ways of exposure to antimicrobials, resistant bacteria, and antimicrobial resistance genes for humans and animals” has recently been published in Spain, within the national program of surveillance of resistance to antibiotics. It emphasizes the relevance of livestock production as a source of emission of antibiotic substances, and its related Public Health concerns, as well as highlighting the need to continue working on the One Health program.
We have added a comment on the subject in the discussion section.
- The authors state that information regarding the antimicrobial use in pig farms was not available. Despite this limitation, can all of the antimicrobials tested in vitro be used in pig production? It would be interesting if this issue could be discussed.
This has been included in the discussion section.
- The authors suggest increased control and surveillance of multiresistant bacteria in pig livestock. Considering that every country of the EU should monitor the antimicrobial resistance in bacteria from livestock, what additional measures should be enforced? It would be interesting if this issue could be discussed.
This has been included in the discussion section.
Reviewer 2 Report
Dear Authors, the presented study concerns an interesting problem related to the prevalence of antibiotic resistant bacteria in animals. However, in order to make your manuscript acceptable for an international recognized journal, some major work needs to be done.
First of all, the quality of English language used needs improvement. Although it is not very poor and the manuscript is understandable, there are many issues related to grammar, synthax or improper use of nouns/verbs.
Then, a lot of details are missing and/or not described properly in the paper.
Some detailed comments can be found below:
Abstract
- 18 – the samples of what exactly were collected? Meat? From what region of the body?
- 22 – how was the presence of these groups investigated?
- 25 - Sequence Type 398 (ST39) it does not mean anything specific while reading the Abstract.
Introduction
- 36-37: this sentence is a cliché and does not mean anything. Please elaborate what do the Authors mean by stating that antibiotic resistance has been linked to humans? to animals? to environmental factors? This is definitely too little to say about antibiotic resistance.
- 37-39. This also needs clarification.
- 80. What does “pig samples” mean? The samples collected from pigs can be of a variety of origin and this needs to be specified at this stage of the paper.
Materials and Methods
- 94 – this is the first place where the reader finds out where the examined samples came from. In my opinion this is too late.
Isolation and Identification
Why only nasal swabs were inoculated on MRSA agar while the rectal swabs were inoculated on other agars? Moreover, information provided in this section is unspecific and the reader does not know what types of bacteria were identified on what media. This needs to be clearly stated in the first paragraph of this section.
Why do the Authors provide detailed information about the MLST-based allelic profile determination within the strains of bacteria that were not detected in the study?
Results
Table 2 – what does SRM mean? Number of examined strains should be provided within the table, which needs to be stand-alone..
Also, the presentation of results needs improvement – you have quite large database and presentation of the results in the form of three tables is not enough.
Discussion
Actually, the comparison of the results obtained in your study with the results obtained by other authors does not fall into the definition of a scientific discussion of the results in an international peer-review journal. Please elaborate more on what your results mean, what may stand behind such observations and what are the consequences as well as possible measures to counteract the spread of antibiotic resistance among animals, humans and the environment.
Author Response
Thank you very much for the comments and time spent reading this work.
Our paper has been translated and revised by a qualified translator, and we hope grammar and syntax have been improved.
Abstract
- 18 – the samples of what exactly were collected? Meat? From what region of the body?
You are right. The origin of the samples was not specified in the Abstract. We have included it now.
22 – how was the presence of these groups investigated?
In the text we have added everything regarding the genetic characterization and antibiogram performed.
- 25 - Sequence Type 398 (ST39) it does not mean anything specific while reading the Abstract.
You are right. We have eliminated this sentence from the abstract and it will be mentioned in the results section.
Introduction
- 36-37: this sentence is a cliché and does not mean anything. Please elaborate what do the Authors mean by stating that antibiotic resistance has been linked to humans? to animals? to environmental factors? This is definitely too little to say about antibiotic resistance.
We have included our opinion on the matter and how the animal reservoir can contribute to the increase of resistances in humans in the discussion section.
- 37-39. This also needs clarification.
It is a quoted sentence from other authors, as we have indicated in the introduction. Human, animal, and environmental health are intrinsically linked forming One Single Health. All action plans against antimicrobial resistances should be approached from this point of view to prevent the emergence of multiresistant strains, also called superbugs. This is also mentioned in the discussion section.
- 80. What does “pig samples” mean? The samples collected from pigs can be of a variety of origin and this needs to be specified at this stage of the paper.
You are right. We have already modified this point in the abstract, and we have eliminated the word sample in this section
Materials and Methods
- 94 – this is the first place where the reader finds out where the examined samples came from. In my opinion this is too late.
Following your indications, we have already included it in the abstract section
Isolation and Identification
Why only nasal swabs were inoculated on MRSA agar while the rectal swabs were inoculated on other agars? Moreover, information provided in this section is unspecific and the reader does not know what types of bacteria were identified on what media. This needs to be clearly stated in the first paragraph of this section.
In the case of Staphylococcus spp., nasal samples were collected since most studies, on animals and humans, indicate that nasal samples are the best to isolate these bacteria. However, for the remaining bacteria studied, rectal samples were used
Following your indications, we have added a clarifying paragraph on that matter in the abstract, material and methods, and discussion sections.
Why do the Authors provide detailed information about the MLST-based allelic profile determination within the strains of bacteria that were not detected in the study?
We have described the whole technique although it was not completed, since all microorganisms were not isolated in the selective culture media. If you so consider we could eliminate the corresponding paragraphs.
Results
Table 2 – what does SRM mean? Number of examined strains should be provided within the table, which needs to be stand-alone.
SRM: This is an error, it has already been modified in the text (MRS). MRS means methicillin-resistant Staphylococcus, including MRSA and MSCoNS.
Following your indications, we have added the number of strains examined in Table 2, making it clearer.
Also, the presentation of results needs improvement – you have quite large database and presentation of the results in the form of three tables is not enough.
We have included the definition of multiresistant strain in the material and methods section, and we have added the percentage of multiresistant strains in the results section, thus broadening the information provided in this section
Discussion
Actually, the comparison of the results obtained in your study with the results obtained by other authors does not fall into the definition of a scientific discussion of the results in an international peer-review journal. Please elaborate more on what your results mean, what may stand behind such observations and what are the consequences as well as possible measures to counteract the spread of antibiotic resistance among animals, humans and the environment.
Thank you for the comment. We agree with you, and we have improved this section according to your indications.
Round 2
Reviewer 1 Report
The authors have revised the manuscript addressing the issues which were pointed out and answered the questions.
Though some points regarding the role of pork, or the incidence of LA-MRSA CC3398, which could contribute greatly to better understand the impact of pig production systems on human health in the Spanish scenario could have been further discussed in the revised version of the manuscript, it is my opinion that the manuscript is now suitable for publication.
Reviewer 2 Report
Dear Authors,
Thank you for submitting the corrected version of the manuscript.
All comments have been addressed.